# Continuous Deep Q-Learning in Optimal Control Problems: Normalized Advantage Functions Analysis

## Abstract

One of the most effective continuous deep reinforcement learning algorithms is normalized advantage functions (NAF). The main idea of NAF consists in the approximation of the Q-function by functions quadratic with respect to the action variable. This idea allows to apply the algorithm to continuous reinforcement learning problems, but on the other hand, it brings up the question of classes of problems in which this approximation is acceptable. The presented paper describes one such class. We consider reinforcement learning problems obtained by the discretization of certain optimal control problems. Based on the idea of NAF, we present a new family of quadratic functions and prove its suitable approximation properties. Taking these properties into account, we provide several ways to improve NAF. The experimental results confirm the efficiency of our improvements.

## 1 Introduction

The standard reinforcement learning (RL) setup consists of an agent interacting with an environment (Sutton & Barto, 2018). At each step of the interaction, the agent determines an action based on its policy and its current state, gets a reward, and makes a transition to the next state. An aim of the agent is to learn the policy that maximizes the sum of rewards.

Q-learning (Watkins & Dayan, 1992) is one of the most widespread algorithms for solving RL problems. According to this algorithm, the optimal action-value function (Q-function) is being found as a solution of the Bellman optimality equation. After the learning, the agent can act optimally by the learned Q-function. Initially, the Q-learning algorithm was applied for solving RL problems with finite state and action spaces. In this case, the Q-function can be represented by a finite table. For the case of a large or continuous state space, Q-learning has recently been extended to Deep Q-learning algorithm (Mnih et al., 2015) that allows to look for the approximate Q-function in the class of neural networks by means of the stochastic gradient descent. Deep Q-learning and its modifications have shown the efficiency for a range of challenging tasks (Wang et al., 2016; van Hasselt et al., 2016; Schaul et al., 2016; Hessel et al., 2018), however, note that this algorithm can not be directly applied for solving RL problems with continuous action spaces. The reason is that Deep Q-learning involves a maximizing of an approximate Q-function by the action variable on each step of the learning, which is a complex problem for continuous action spaces. Among various approaches to overcome this problem (Lillicrap et al., 2016; Haarnoja et al., 2017; Kalashnikov et al., 2018; Lim et al., 2019; Ryu et al., 2020; Lutter et al., 2021), we focus on an idea of the normalized advantage functions (NAF) algorithm (Gu et al., 2016). This idea consists in the approximation of the Q-function by functions quadratic with respect to the action variable. It allows to get the maximum quite fast and precisely and solve some challenging control problems (Gu et al., 2017; Dong et al., 2018; Ikemoto & Ushio, 2021), but on the other hand, it brings up the question of classes of RL problems in which this approximation is acceptable. The presented paper describes one of the possible answers to this question.

Note that the class of LQR problems (Bradtke et al., 1994) has the Q-functions quadratic with respect to the action variable. However, this class, being quite special, is not suitable for the description of complex controlled processes. In the paper, we consider a wider (in some sense) class of RL prob-

lems. We consider RL problems which are obtained by the discretization of certain optimal control problems (Bardi & Dolcetta, 1997). The rationale for the consideration of such RL problems is that a lot of RL problems with continuous action spaces arise from control problems for mechanical or robotic systems (Lillicrap et al., 2016; Gu et al., 2016; Haarnoja et al., 2017; Gu et al., 2017; Kalashnikov et al., 2018), whose dynamic are, in fact, described by ordinary differential equations.

For the considered class of problems, based on the idea of NAF, we present a new family of quadratic functions and prove that, first, this family is sufficiently rich to approximately solve the Bellman optimality equation (Theorem 1), and second, any sufficiently accurate solution of the Bellman optimality equation allows to approximately obtain the optimal policy in the corresponding optimal control problem (Theorem 2). Moreover, we prove that it is impossible to get the same results for the original family of functions from Gu et al. (2016) (Theorem 3). From the obtained theoretical statements, we get some additional knowledge about the Q-function approximation by our family of quadratic functions and also provide several ways to use this knowledge in order to improve NAF. The experimental results confirm the efficiency of our improvements.

## 2 BACKGROUND

The standard reinforcement learning (RL) setup consists of an agent interacting with an environment (Sutton & Barto, 2018). This interaction is described by a Markov Decision Process (MDP), which is a tuple $(\mathcal{S}, \mathcal{U}, \mathcal{P}, \mathcal{R}, \rho_0, \gamma)$, where $\mathcal{S}$ is a state space, $\mathcal{U}$ is an action space, $\mathcal{P}(s'|s, u)$ is a transition distribution, $\mathcal{R}(s, u)$ is a reward function, $\rho_0(s)$ is an initial state distribution, and $\gamma \in [0, 1]$ is a discount factor. An aim of the agent is to learn its optimal policy $\mu_*(s)$ that maximizes the value

$$J(\mu) = \mathbb{E}\left[\sum_{i=0}^{\infty} \gamma^i \mathcal{R}(s_i, u_i) \,|\, s_0 \sim \rho_0(s_0),\, u_i = \mu(s_i),\, s_{i+1} \sim \mathcal{P}(s_{i+1}|s_i, u_i), i = 0, 1, 2, \dots\right].$$

In the general statement of reinforcement learning problems, a policy of the agent can be stochastic, however, within this paper, we assume that the policy is deterministic.

One of the most effective algorithms for solving RL problems is Q-learning (Watkins & Dayan, 1992). According to this algorithm, the agent explores the environment and looks for the optimal action-value function (Q-function)

$$Q_*(s, u) = \sup_{\mu} \mathbb{E}\left[\sum_{i=0}^{\infty} \gamma^i \mathcal{R}(s_i, u_i) \,|\, s_0 = s,\, u_0 = u,\right.$$
$$\left. s_{i+1} \sim \mathcal{P}(s_{i+1}|s_i, u_i),\, u_{i+1} = \mu(s_{i+1}),\, i = 0, 1, 2, \dots\right].$$

as a solution of the Bellman optimality equation

$$Q_*(s, u) = \mathbb{E}\left[\mathcal{R}(s, u) + \gamma \max_{u' \in \mathcal{U}} Q_*(s', u') \,|\, s' \sim P(s'|s, u)\right].$$

In other words, the agent solves the following minimization problem:

$$\sup_{s \in \mathcal{S}, u \in \mathcal{U}} \left| Q(s, u) - \mathbb{E}\left[\mathcal{R}(s, u) + \gamma \max_{u' \in \mathcal{U}} Q(s', u') \,|\, s' \sim \mathcal{P}(s'|s, u)\right] \right| \to \inf_{Q}. \quad (1)$$

If the agent knows the function $Q_*(s, a)$, it can act optimally by the greedy policy $\mu_*(s) \in \arg\max_{u \in \mathcal{U}} Q_*(s, u)$.

Initially, the Q-learning algorithm was applied for solving RL problems with finite state and action spaces. In this case, the Q-function is represented by a finite table and problem (1) is finite-dimensional. For the case of a large or continuous state space, Q-learning has recently been extended to Deep Q-learning algorithm (Mnih et al., 2015) that allows to look for the approximate Q-function in the class of neural networks $Q(x, u|\theta^Q)$, where $\theta^Q$ is the parameter vector of the neural network. During the learning, the experiences $(s_i, u_i, r_i, s_{i+1})$ are stored in the buffer $\mathcal{D}$ and simultaneously the parameter vector $\theta^Q$ is updated by means of the stochastic gradient descent minimizing the loss function

$$L(\theta^Q) = \mathbb{E}\left[(Q(s, u|\theta^Q) - y)^2 \,|\, (s, u, r, s') \sim U(\mathcal{D})\right], \quad y = r + \gamma \max_{u' \in \mathcal{U}} Q(s', u'|\theta^Q). \quad (2)$$

where $U(\mathcal{D})$ is the uniform distribution on $\mathcal{D}$. Deep Q-learning and its modifications are effective for a range of challenging tasks (Wang et al., 2016; van Hasselt et al., 2016; Schaul et al., 2016; Hessel et al., 2018), however, note that this algorithm can not be directly applied for solving RL problems with continuous action spaces. The reason is that Deep Q-learning involves the maximizing in (2) on each step of the learning, which is a complex problem for continuous $\mathcal{U}$. Among various approaches to overcome this problem (Lillicrap et al., 2016; Haarnoja et al., 2017; Kalashnikov et al., 2018; Lim et al., 2019; Ryu et al., 2020; Lutter et al., 2021), we focus on an idea of the normalized advantage functions (NAF) algorithm (Gu et al., 2016). This idea consists in the approximation of the Q-function by the following quadratic with respect to $u$ functions:

$$Q(s, u|\theta^Q) = V(s|\theta^V) + A(s, u|\theta^A),$$
$$A(s, u|\theta^A) = -\frac{1}{2}(u - \mu(s|\theta^\mu))^T P(s|\theta^P)(u - \mu(s|\theta^\mu)), \tag{3}$$

where $V(s|\theta^V)$, $\mu(s|\theta^\mu)$, and $P(s|\theta^P)$ are neural networks with parameters $\theta^V$, $\theta^\mu$, and $\theta^P$, respectively; $P(s|\theta^P)$ is a positive-definite square matrix for each $s$ and $\theta^P$; $\theta^A = \{\theta^\mu, \theta^P\}$ and $\theta^Q = \{\theta^A, \theta^V\}$. Under the condition

$$\mu(s|\theta^\mu) \in U, \tag{4}$$

it allows to get the maximum and argmaximum values directly by values of $V(s|\theta^V)$ and $\mu(s|\theta^\mu)$:

$$\max_{u \in \mathcal{U}} Q(s, u|\theta^Q) = V(s|\theta^V), \quad \underset{u \in \mathcal{U}}{\text{Argmax}}\, Q(s, u|\theta^Q) = \mu(s|\theta^\mu), \tag{5}$$

but on the other hand, it brings up the question of classes of RL problems in which quadratic approximations is acceptable. Below, we describes one such class.

## 3   PROBLEM STATEMENT

In this section, we consider a certain class of optimal control problems and show that discrete approximations of these problems can be formalized as RL problems.

Consider the following optimal control problem: it is required to maximize the functional

$$J(u(\cdot)) = \sigma(x(T)) - \int_0^T \big(q(t, x(t)) + u(t)^T r(t, x(t))u(t)\big)dt, \tag{6}$$

over all $u(\cdot)$, where $x(\cdot)$ is the solution (Filippov, 1988, §1) of the differential equation

$$\frac{d}{dt}x(t) = f(t, x(t)) + g(t, x(t))u(t), \quad t \in [0, T], \tag{7}$$

under the initial condition

$$x(0) = z. \tag{8}$$

Here $t$ is the time variable, $T > 0$ is the terminal instant of time, $x(t) \in \mathbb{R}^n$ is the current state vector, $u(t) \in U$ is the current control action vector forming the measurable function $u(\cdot)$, $U \subset \mathbb{R}^m$ is the nonempty compact set, $z \in \mathbb{R}^n$ is the fixed initial state vector, $f(t, x) \in \mathbb{R}^n$, $g(t, x) \in \mathbb{R}^{n \times m}$, $q(t, x) \in \mathbb{R}$, $r(t, x) \in \mathbb{R}^{m \times m}$, $(t, x) \in [0, T] \times \mathbb{R}^n$ are continuous with respect to $t$ and continuously differentiable with respect to $x$ functions, $r(t, x)$ is the positive-definite matrix for each $(t, x) \in [0, T] \times \mathbb{R}^n$, and $\sigma(x) \in \mathbb{R}$, $x \in \mathbb{R}^n$ is the continuous function. We assume that there exists a constant $c_{fg} > 0$ such that

$$\|f(t, x) + g(t, x)u\| \le (1 + \|x\|)c_{fg}, \quad (t, x) \in [0, T] \times \mathbb{R}^n, \quad u \in U. \tag{9}$$

Note that, under these conditions, for each function $u(\cdot)$, there exists a unique solution $x(\cdot)$ of equation (7) under the initial condition (8) (Filippov, 1988, §1).

Define the value function in optimal control problem (6), (7) by

$$V_*(t_*, x_*) = \sup_{u(\cdot)} \left(\sigma(x(T)) - \int_{t_*}^T \big(q(t, x(t)) + u(t)^T r(t, x(t))u(t)\big)dt\right), \quad (t_*, x_*) \in [0, T] \times \mathbb{R}^n, \tag{10}$$

where, for each $u(\cdot)$, $x(\cdot)$ is the solution of equation (7) on the interval $[t_*, T]$ under the initial condition $x(t_*) = x_*$.

Define the sets

$$S = \big\{(t,x) \in [0,T] \times \mathbb{R}^n \colon \|x\| \leq (1 + \|z\|)e^{c_{fg}t} - 1\big\}, \quad S(t) = \big\{x \in \mathbb{R}^n \colon (t,x) \in S\big\}. \quad (11)$$

Let $k \in \mathbb{N}$, $\Delta t_k = T/k$, and $t_i = i\Delta t_k$, $i \in \overline{0,k}$. Consider the corresponding discrete optimal control problem: it is required to maximize the function

$$J_k(u_0, u_1, \ldots u_{k-1}) = \sigma(x_k) - \Delta t_k \sum_{i=0}^{k-1} \big(q(t_i, x_i) + u_i^T r(t_i, x_i)u_i\big), \quad (12)$$

over all $u_i \in U$, $i \in \overline{0, k-1}$, where $(x_0, x_1, \ldots, x_k)$ is defined by

$$x_0 = z, \quad x_{i+1} = x_i + (f(t_i, x_i) + g(t_i, x_i)u_i)\Delta t_k, \quad i \in \overline{0, k-1}. \quad (13)$$

Let us show that problem (12), (13) can be formalized as the RL problem. First, we define the state and actions spaces, the initial state distribution, and the discount factor as follows:

$$\mathcal{S} = \cup_{i=0}^k \big(\{t_i\} \times S(t_i)\big) \cup s_T, \quad \mathcal{U} = U, \quad \rho_0(s_0) = \delta(s_0 = (0,z)), \quad \gamma = 1. \quad (14)$$

Here $s_T$ is some fictional terminal state, $\delta$ is Dirac delta distribution. Next, for every $i \in \overline{0, k-1}$, $x \in S(t_i)$, and $u \in U$, we define the transition distribution and the reward function by

$$\mathcal{P}(s'|s = (t_i, x), u) = \delta(s' = (t_{i+1}, x')), \quad \mathcal{R}(s = (t_i, x), u) = -\big(q(t_i, x) + u^T r(t_i, x)u\big)\Delta t_k, \quad (15)$$

where $x' = x + (f(t_i, x) + g(t_i, x)u)\Delta t_k$. Taking into account (9) and (11), one can prove the inclusion $(t_{i+1}, x') \in \mathcal{S}$. Hence, the transition distribution $\mathcal{P}$ is well-defined. For $i = k$, we set

$$\mathcal{P}(s'|s = (t_k, x), u) = \delta(s' = s_T), \quad \mathcal{R}(s = (t_k, x), u) = \sigma(x), \quad x \in S(t_k), \quad u \in U. \quad (16)$$

In order to make dynamical processes (13) formally infinite, we put

$$\mathcal{P}(s'|s_T, u) = \delta(s' = s_T), \quad \mathcal{R}(s_T, u) = 0, \quad u \in U. \quad (17)$$

Thus, we define MDP which describes the RL problem corresponding to problem (12), (13). Next, we show that such RL problems is suitable for using quadratic approximations of the Q-function.

## 4 QUADRATIC APPROXIMATIONS OF THE Q-FUNCTION

Denote by $\mathbb{Q}$ the family of functions $Q$ such that

$$Q(t, x, u) = V(t, x) + A(t, x, u), \quad (t, x, u) \in [0, T) \times \mathbb{R}^n \times U, \quad (18)$$

where

$$A(t, x, u) = -(u - \tilde{\mu}(t, x))^T P(t, x)(u - \tilde{\mu}(t, x))$$
$$+ (\mu(t, x) - \tilde{\mu}(t, x))^T P(t, x)(\mu(t, x) - \tilde{\mu}(t, x)) \quad (19)$$
$$\mu(t, x) \in \arg\min_{u' \in U} (u' - \tilde{\mu}(t, x))^T P(t, x)(u' - \tilde{\mu}(t, x)).$$

Here $V(t, x)$, $\tilde{\mu}(t, x)$, and $P(t, x)$ are continuous functions; $P(t, x)$ is a positive-definite square matrix for each $(t, x) \in [0, T) \times \mathbb{R}^n$.

Note that, functions $Q$ from the family $\mathbb{Q}$ satisfy the equalities

$$\max_{u \in U} Q(t, x, u) = V(t, x), \quad \operatorname*{Argmax}_{u \in U} Q(t, x, u) = \mu(t, x), \quad (20)$$

as well as (see (5)) quadratic functions from family (3). However, these function families are different. The difference is that we do not assume the inclusion $\tilde{\mu}(t, x) \in U$ as opposed to assumption (4).

The theorems below establish a connection between the optimal control problem (6)–(8) and minimization problems (1) for MDP (14)–(17).

**Theorem 1.** Let the value function $V_*(t, x)$ be continuously differentiable. Then, for every $\varepsilon > 0$, there exists $k_* > 0$ such that, for every $k \geq k_*$, the function $Q \in \mathbb{Q}$ defined by (18) and (19) where

$$V(t, x) = V_*(t, x), \quad P(t, x) = r(t, x)\Delta t_k, \quad \tilde{\mu}(t, x) = \frac{1}{2}r^{-1}(t, x)g^T(t, x)\nabla_x V_*(t, x) \quad (21)$$

satisfies the inequality

$$\left| Q(t_i, x, u) + \left( q(t_i, x) + u^T r(t_i, x)u \right)\Delta t_k - \max_{u' \in U} Q(t_{i+1}, x', u') \right| \leq \varepsilon \Delta t_k,$$

$$x' = x + \left( f(t_i, x) + g(t_i, x)u \right)\Delta t_k, \quad u \in U, \quad x \in S(t_i), \quad i \in \overline{0, k-1}, \quad (22)$$

where we assume $Q(t_k, x', u') = \sigma(x')$.

**Theorem 2.** Let the value function $V_*(t, x)$ be continuously differentiable. Let $\varepsilon > 0$ and $k_* > 0$ be defined according to Theorem 1. Take $k \geq k_*$ and suppose that a function $Q \in \mathbb{Q}$ satisfies inequality (22). Then the following estimate holds:

$$J_k(u_0, u_1, \ldots u_{k-1}) \geq \sup_{u(\cdot)} J(u(\cdot)) - 3T\varepsilon, \quad (23)$$

where the function $J_k(u_0, u_1, \ldots u_{k-1})$ is defined by (12) with $u_i = \mu(t_i, x_i)$, $i \in \overline{0, k-1}$ and the function $\mu(t, x)$ is defined by $Q$ according to (18) and (19).

Thus, Theorem 1 shows that the function family $\mathbb{Q}$ is sufficiently rich to contain approximate solutions of problem (1) with a predetermined accuracy and Theorem 2 establishes that all such approximate solutions contained in $\mathbb{Q}$ allow to get the policy, which approximately provides the optimal result in optimal control problem (6)–(8).

Note that the similar results can be obtained for the wider class of optimal control problems (see Appendix B). However, in this case, we need to consider another form of the function $A(t, x, u)$. Within the presented paper, we focus on class (6)–(8), because, firstly. this class seems quite general and important for applications and, secondly, the function $A(t, x, u)$, corresponding to this class, has the quite simple (quadratic) form.

Now, let us consider the original family of functions from Gu et al. (2016). Denote by $\mathbb{Q}_{NAF}$ the family of functions $Q$ such that

$$Q(t, x, u) = V(t, x) - (u - \mu(t, x))^T P(t, x)(u - \mu(t, x)), \quad (t, x, u) \in [0, T] \times \mathbb{R}^n \times U, \quad (24)$$

where $V(t, x)$, $\mu(t, x)$, and $P(t, x)$ are continuous functions; $P(t, x)$ is a positive-definite square matrix for each $(t, x) \in [0, T] \times \mathbb{R}^n$; $\mu(t, x) \in U$ for each $(t, x) \in [0, T] \times \mathbb{R}^n$. The theorem below establishes that if we take the family $\mathbb{Q}_{NAF}$ instant of $\mathbb{Q}$, then Theorem 1 can not be proved even in the simplest cases of optimal control problem (6)–(8).

**Theorem 3.** Let $n = m = 1$, $T = 1$, $U = [-1, 1]$, $f(t, x) = q(t, x) = 0$, $g(t, x) = r(t, x) = 1$, $\sigma(x) = -x^2$, and $z = 2$. Then, for every $k \geq 4$ and $Q \in \mathbb{Q}_{NAF}$, there exist $i \in \overline{0, k-1}$, $x \in S(t_i)$, and $u \in U$ such that

$$\left| Q(t_i, x, u) + u^2 \Delta t_k - \max_{u' \in U} Q(t_{i+1}, x', u') \right| > \Delta t_k/8, \quad x' = x + u\Delta t_k,$$

where we assume $Q(t_k, x', u') = -(x')^2$.

Proofs of the theorems are given in the Appendix A.

## 5 EXPERIMENTS

We consider four examples of optimal control problems (6)–(8) described in Table 1, where

$$\sigma_1(x) = -x_1^2 - x_2^2, \quad \sigma_2(x) = -|x_1| - 0.1|x_2|, \quad \sigma_3(x) = -|x_1 - 4| - |x_2| - |x_3 - 0.75\pi|$$

$$\sigma_3(x) = -x_1^2 - x_2^2 - (x_3 - 2)^2 - (x_4 - 2)^2$$

Van der Pol oscillator is a famous model of a non-conservative oscillator with non-linear damping. The aim of the control is to stabilize the oscillator at the terminal time.

Table 1: Parameters in the examples of optimal control problems

| Name | $n$ | $m$ | $T$ | $U$ | $f(t,x)$ | $g(t,x)$ | $q(t,x)$ | $r(t,x)$ | $\sigma$ | $z$ |
|---|---|---|---|---|---|---|---|---|---|---|
| Van der Pol oscillator | 2 | 1 | 11 | $[-1,1]$ | $\begin{pmatrix} x_2 \\ (1-x_1^2)x_2 \end{pmatrix}$ | $\begin{pmatrix} 0 \\ -x_1 \end{pmatrix}$ | 0 | 0.05 | $\sigma_1(x)$ | $\begin{pmatrix} 1 \\ 0 \end{pmatrix}$ |
| Pendulum | 2 | 1 | 5 | $[-2,2]$ | $\begin{pmatrix} x_2 \\ 14.7\sin(x_1) \end{pmatrix}$ | $\begin{pmatrix} 0 \\ 3 \end{pmatrix}$ | 0 | 0.01 | $\sigma_2(x)$ | $\begin{pmatrix} \pi \\ 0 \end{pmatrix}$ |
| Dubins car | 3 | 1 | $2\pi$ | $[-0.5,1]$ | $\begin{pmatrix} \cos(x_3) \\ \sin(x_3) \\ 0 \end{pmatrix}$ | $\begin{pmatrix} 0 \\ 0 \\ 1 \end{pmatrix}$ | 0 | 0.05 | $\sigma_3(x)$ | $\begin{pmatrix} 0 \\ 0 \\ 0 \end{pmatrix}$ |
| A target problem | 6 | 2 | 10 | $[-1,1]^2$ | $\begin{pmatrix} 0 \\ 0 \\ x_5 \\ x_6 \\ x_1-x_3 \\ x_2-x_4 \end{pmatrix}$ | $\begin{pmatrix} 1 & 0 \\ 0 & 1 \\ 0 & 0 \\ 0 & 0 \\ 0 & 0 \\ 0 & 0 \end{pmatrix}$ | 0 | 0.001 | $\sigma_4(x)$ | $\begin{pmatrix} 0 \\ 0 \\ 0 \\ 0 \\ 0 \\ 0 \end{pmatrix}$ |

Pendulum is a traditional problem for testing control algorithms. The aim of the control is the stabilization of the pendulum in the top position at the terminal time.

Dubins car is a quite famous model which describes a motion of the point particle moving at a constant speed on the plane. The problem is to find a control providing the closeness of the motion with a target point at the terminal time.

A target problem is an optimal control problem presented in Munos (2006). The dynamic system describes a hand holding a spring to which is attached a mass. It is required to control the hand such that the mass achieve the target point at the terminal time.

## 5.1 BOUNDED NAF

First, we modify NAF algorithm, proposed in Gu et al. (2016), based on the function family $\mathbb{Q}$. Note that, the considered examples have $U = [\alpha, \beta]^m$. Denote $\tanh_{\alpha,\beta}(\nu) = \alpha + (1 + \tanh(\nu))(\beta - \alpha)/2$, $\nu \in \mathbb{R}^m$, $\alpha, \beta \in \mathbb{R}^m$, where $\tanh$ is the hyperbolic tangent for each coordinate. Then, according to (18) and (19), we can use the following approximation of the Q-function, within NAF algorithm:

$$Q(t,x,u|\theta^Q) = V(t,x|\theta^V) + A(t,x,u|\theta^A),$$

$$A(t,x,u|\theta^A) = -\big(u - \tilde{\mu}(t,x|\theta^{\tilde{\mu}})\big)^2 P(t,x|\theta^P)$$

$$+\big(\tanh_{\alpha,\beta}(\tilde{\mu}(t,x|\theta^{\tilde{\mu}})) - \tilde{\mu}(t,x|\theta^{\tilde{\mu}})\big)^2 P(t,x|\theta^P),$$

where $V(t,x|\theta^V)$, $\tilde{\mu}(t,x|\theta^{\tilde{\mu}})$, and $P(t,x|\theta^P)$ are neural networks with parameters $\theta^V$, $\theta^{\tilde{\mu}}$, and $\theta^P$, respectively; $P(t,x|\theta^P)$ is a positive-definite square matrix for each $(t,x)$ and $\theta^P$; $\theta^A = \{\theta^{\tilde{\mu}}, \theta^P\}$ and $\theta^Q = \{\theta^A, \theta^V\}$. To be short, we call this algorithm Bounded NAF (BNAF), because it is essential for our modification of NAF that the set $U$ is bounded.

Note that, we can also use the function $\mathrm{clip}_{\alpha,\beta}(\nu) = \max\{\alpha, \min\{\beta, \nu\}\}$, $\nu \in \mathbb{R}^m$, $\alpha, \beta \in \mathbb{R}^m$ instead of the function $\tanh_{\alpha,\beta}$. According to (19), it is more correct, however our experiments show that the learning results are slightly better with $\tanh_{\alpha,\beta}$ (see Appendix C). A possible reason for this is that the function $\mathrm{clip}_{\alpha,\beta}$ is not smooth.

## 5.2 REWARD-BASED BNAF

If we know the function $r(t,x)$, then, according to Theorem 1, we can use the function $r(t,x)\Delta t_k$ instead of the neural network $P(t,x|\theta^P)$ to reduce the number of learning parameters. This variant of BNAF is called Reward-based BNAF (RB-BNAF).

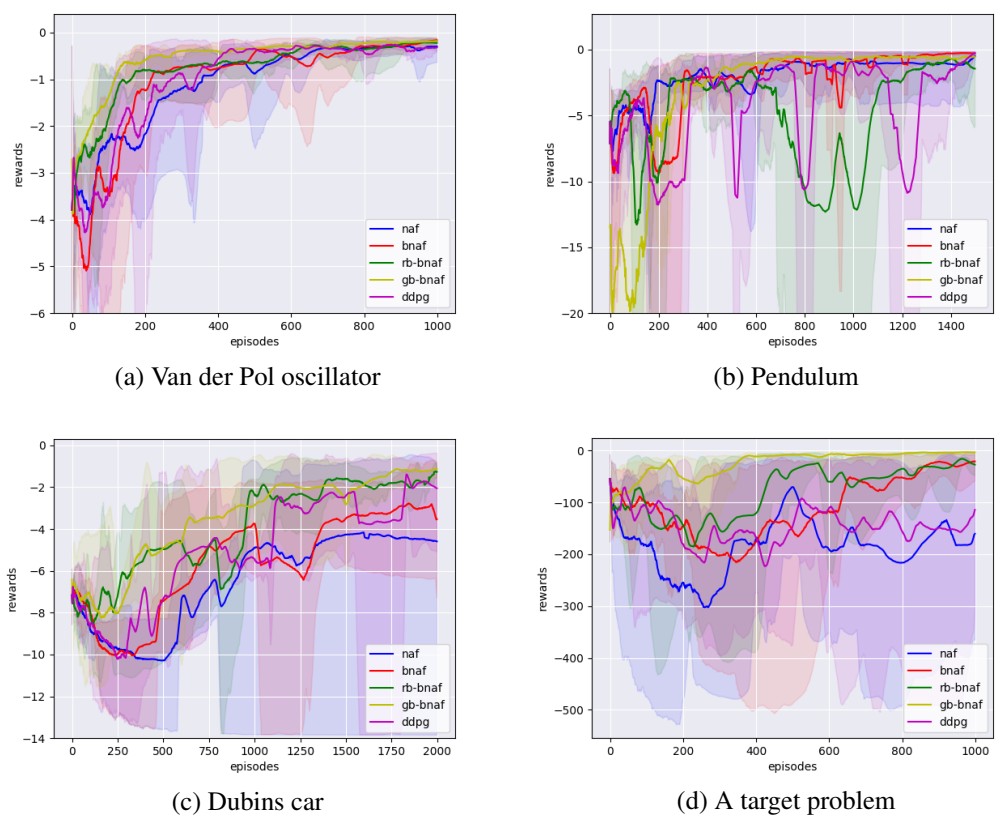

(a) Van der Pol oscillator

(b) Pendulum

(c) Dubins car

(d) A target problem

Figure 1: Results of NAF, BNAF, RB-BNAF, GB-BNAF, DDPG algorithms averaged over 5 seeds.

## 5.3 GRADIENT-BASED BNAF

If we also know the function $g(t, x)$, then, according to the Theorem 1, we can use the function

$$\tilde{\mu}(t, x|\theta^V) = \frac{1}{2} r^{-1}(t, x) g^T(t, x) \nabla_x V(t, x|\theta^V)$$

instead of the neural network $\tilde{\mu}(t, x|\theta^{\tilde{\mu}})$. It also reduces the number of learning parameters. This variant of BNAF is called Gradient-based BNAF (GB-BNAF).

## 5.4 EXPERIMENTAL RESULTS

We use the same learning parameters of every our tasks. We apply neural networks with two layers of 256 and 128 rectified linear units (ReLU) and learn their used ADAM with the learning rate $lr = 5e^{-4}$. We use batch size $n_{bs} = 256$ and smoothing parameter $\tau = 1e^{-3}$. Also we take $\Delta t = 0.1$. All calculations were performed on a personal computer in a standard way.

We compere NAF, BNAF, RB-BNAF, GB-BNAF algorithms, and DDPG algorithm presented in (Lillicrap et al., 2016). Figure 1 shows learning curves of the algorithms for the considered examples. One can note that GB-BNAF algorithm is the most stable and gets the best performance in all examples. RB-BNAF is also capable to get acceptable results in all examples, although it does not have the same stable learning as GB-BNAF. RB-BNAF algorithm gets worse result than GB-BNAF and RB-BNAF in Dubins car (c). NAF demonstrate good results in Van der Pol oscillator (a) and Pendulum (b), however it does not cope with Dubins car (c) and A target problem (d). DDPG algorithm shows poor performance only in A target problem (d).

Thus, taking the presented experiments into account, we can give the following general recommendations. For solving optimal control problems (6)–(8), it is rational to use BNAF, GB-BNAF, and

especially RB-BNAF algorithms along with NAF algorithm. There is a high probability that they will show better results and more stable learning.

## 6 RELATED WORKS

Many different ways to apply reinforcement learning approaches for solving optimal control problems are investigated (Baird, 1994; Doya, 1995; 2000; Munos, 2006; Tallec et al., 2019; Jeongho Kim, 2020; Lutter et al., 2021). Among them, a time-discretization is perhaps the most obvious and widely used tool (Lillicrap et al., 2016; Gu et al., 2016; Haarnoja et al., 2017; Gu et al., 2017; Kalashnikov et al., 2018). From the theoretical point of view, it is known (Bardi & Dolcetta, 1997, p.388) that solutions of time-discrete optimal control problems converge to a solution of the initial problem as the discretization step tends to zero. In the present paper, we also use the time-discretization and study approximating solutions of Bellman equations and the corresponding greedy politics depending on the discretization step (see Theorem 1 and 2). Other studies of dependencies on the discretization step of reinforcement learning methods can be found in Munos (2006); Tallec et al. (2019).

We focus on the idea from Gu et al. (2016) to expand the Q-learning algorithm to optimal control and reinforcement learning problems with continuous actions. Other approaches for solving such problems are investigated in Lillicrap et al. (2016); Haarnoja et al. (2017); Kalashnikov et al. (2018); Lim et al. (2019); Ryu et al. (2020); Lutter et al. (2021). The paper Lutter et al. (2021) seems the closest to the presented paper. In this paper, the similar optimal control problem and feedback control policy are considered, however, another Bellman equation is used.

Let us also note that the family of functions (18), (19) is included, in some sense, to families of Q-functions considered in Wang et al. (2016); Tallec et al. (2019). It seems expected because more general classes of problems are considered in these papers. Nevertheless, proposed algorithms and results of these papers are very different from presented in this paper.

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

## A APPENDIX

Denote

$$\mathcal{H}(t, x, s, u) = \langle f(t, x) + g(t, x)u, s \rangle - q(t, x) - u^T r(t, x)u. \tag{25}$$

**Lemma 1.** Let the value function $V_*(t, x)$ be continuously differentiable. Then, the following Hamilton-Jacobi-Bellman equation holds:

$$\frac{\partial V_*(t, x)}{\partial t} + \max_{u \in U} \mathcal{H}(t, x, \nabla_x V_*(t, x), u) = 0, \quad (t, x) \in (0, T) \times \mathbb{R}^n.$$

The proof can be found, for example, in J. Yong. Differential Games: A Concise Introduction. World Scientific, University of Central Florida, 238 USA, 2015. doi: 10.1142/9121.

**Proof of Theorem 1.** Note that, due to (19), (21), and (25), we have

$$A(t, x, u)/\Delta t_k = \mathcal{H}(t, x, \nabla_x V_*(t, x), u) - \max_{u' \in U} \mathcal{H}(t, x, \nabla_x V_*(t, x), u') \tag{26}$$

for any $(t, x, u) \in [0, T] \times \mathbb{R}^n \times U$. Due to (9) and (11), we have

$$\|f(t, x) + g(t, x)u\| \le c_{fg}(1 + \|z\|)e^{c_{fg}T} := \alpha_{fg}, \quad (t, x) \in S, \quad u \in U. \tag{27}$$

Let $\varepsilon > 0$. Since $V_*(t, x)$ is continuously differentiable, there exists $\delta > 0$ such that

$$\left| V_*(t', x') - V_*(t, x) - \frac{\partial V_*(t, x)}{\partial t}(t' - t) - \langle \nabla_x V(t, x), x' - x \rangle \right| \le \big( |t - t'| + \|x - x'\| \big) \frac{\varepsilon}{1 + \alpha_{fg}} \tag{28}$$

for any $(t, x), (t', x') \in S$ satisfying $|t - t'| + \|x - x'\| \le \delta$. Put $k_* = (1 + \alpha_{fg})T/\delta$.

Let $k \ge k_*$, $i \in \overline{0, k-1}$, $x \in S(t_i)$, $u \in U$, and $x' = x + (f(t_i, x) + g(t_i, x)u)\Delta t_k$. Due to (9) and (11), we have $(t_{i+1}, x') \in S$. Then, from (27) and (28), we derive

$$\left| V_*(t_{i+1}, x') - V_*(t_i, x) - \frac{\partial V_*(t_i, x)}{\partial t}\Delta t_k - \langle \nabla_x V_*(t_i, x), f(t_i, x) + g(t_i, x)u \rangle \Delta t_k \right| \le \varepsilon \Delta t_k, \tag{29}$$

Next, according to (18), (20), (25), (26), (29), and Lemma 1, we obtain

$$\left| Q(t_i, x, u) + \big( q(t_i, x) + u^T r(t_i, x)u \big)\Delta t_k - \min_{u'} Q(t_{i+1}, x', u') \right|$$

$$= \left| V_*(t_i, x) + A(t_i, x, u) + \big( q(t_i, x) + u^T r(t_i, x)u \big)\Delta t_k - V_*(t_{i+1}, x') \right|$$

$$\le \left| \frac{\partial V_*(t_i, x)}{\partial t} + \max_{u' \in U} \mathcal{H}(t_i, x, \nabla_x V_*(t_i, x), u') \right| \Delta t_k + \varepsilon \Delta t_k = \varepsilon \Delta t_k.$$

The theorem is proved.

**Proof of Theorem 2.** Let $\varepsilon > 0$. Define $k_*$ according to Theorem 1. Let $k \ge k_*$. Let $Q \in \mathbb{Q}$ satisfy (22). Let $x_i$, $i \in \overline{0, k}$ be defined by (13), where $u_i \in \mu(t_i, x_i)$. Then, due to (12), (20), and (22), we have

$$V(0, z) - J_k(u_0, \ldots, u_{k-1})$$

$$= \sum_{i=0}^{k-1} \Big( Q(t_i, x_i, u_i) + \big( q(t_i, x_i) + u_i^T r(t_i, x_i)u_i \big)\Delta t_k - \max_{u' \in U} Q(t_{i+1}, x_{i+1}, u') \Big) \le T\varepsilon. \tag{30}$$

Let us define $Q_* \in \mathbb{Q}$ according to (21). Let $x_i^*$, $i \in \overline{0, k}$ be defined by (13), where $u_i^* \in \arg\max_{u \in U} Q_*(t_i, x_i, u)$. Then, due to the Theorem 1 and (20), we have

$$\left| V_*(t_i, x_i^*) + \big( q(t_i, x_i^*) + (u_i^*)^T r(t_i, x_i^*)u_i^* \big)\Delta t_k - V_*(t_{i+1}, x_{i+1}^*) \right| \le \varepsilon \Delta t_k. \tag{31}$$

Note that, according to definition (10) of the value function $V_*$, we have $V_*(t_k, x) = \sigma(x)$, $x \in \mathbb{R}^n$. Then, from (20), (22), and (31), we derive

$$0 = \max_{u' \in U} Q(t_k, x_k^*, u') - \sigma(x_k^*)$$

$$\le \sum_{i=0}^{k-1} \Big( \max_{u' \in U} Q(t_{i+1}, x_{i+1}^*, u') - Q(t_i, x_i^*, u_i^*) - \big( q(t_i, x_i^*) + (u_i^*)^T r(t_i, x_i^*)u_i^* \big)\Delta t_k \Big) + V(0, z)$$

$$+ \sum_{i=0}^{k-1} \left( V_*(t_i, x_i^*) + \left( q(t_i, x_i^*) + (u_i^*)^T r(t_i, x_i^*) u_i^* \right) \Delta t_k - V_*(t_{i+1}, x_{i+1}^*) \right) - V_*(0, z)$$

$$\leq 2T\varepsilon + V(0, z) - V_*(0, z).$$

Due to this inequality and inequality (30), we obtain $J_k(u_0, \ldots, u_{k-1}) \geq V_*(0, z) - 3T\varepsilon$. Taking into account (6) and (10) we have $\sup_{u(\cdot)} J(u(\cdot)) = V_*(0, z)$. Thus, the theorem is proved.

**Proof of Theorem 3.** Let $k \geq 4$ and $Q \in \mathbb{Q}_{NAF}$. Let us take $i = k - 1$ and $x = 2$. For the sake of brevity, denote $V = V(t_{k-1}, 2)$, $\mu = \mu(t_{k-1}, 2)$, and $P = P(t_{k-1}, 2)$. Then, in order to prove the theorem, we have to show that

$$\max_{u \in [-1, 1]} \left| V - (u - \mu)^2 P + u^2 \Delta t_k + (2 + u\Delta t_k)^2 \right| > \Delta t_k / 8.$$

Arguing by contradiction assume that

$$\max_{u \in [-1, 1]} \left| V - (u - \mu)^2 P + u^2 \Delta t_k + (2 + u\Delta t_k)^2 \right| \leq \Delta t_k / 8.$$

Then, for $u = -1$, $u = 1$, and $u = 0$, we have

$$- \Delta t_k / 8 \leq V - (1 + \mu)^2 P + \Delta t_k + (2 - \Delta t_k)^2 \leq \Delta t_k / 8, \tag{32}$$

$$- \Delta t_k / 8 \leq V - (1 - \mu)^2 P + \Delta t_k + (2 + \Delta t_k)^2 \leq \Delta t_k / 8, \tag{33}$$

$$- \Delta t_k / 8 \leq -V + \mu^2 P - 4 \leq \Delta t_k / 8. \tag{34}$$

Adding up inequations (32), (33), and twice inequity (34), we derive

$$- \Delta t_k / 2 \leq 2P - 2\Delta t_k - 2\Delta t_k^2 \leq \Delta t_k / 2. \tag{35}$$

Adding up twice inequations (32), (34), and inequity (35), we obtain

$$-\Delta t_k \leq -4\mu P - 8\Delta t_k \leq \Delta t_k.$$

From this estimate, taking into account (35) and the inequality $k \geq 4$, we conclude

$$\mu \leq -\frac{7\Delta t_k}{4P} \leq -\frac{7}{5 + 4\Delta t_k} \leq -\frac{7}{6} < -1.$$

This inequality contradicts the inclusion $\mu \in U = [-1, 1]$, which should be valid for $Q \in \mathbb{Q}_{NAF}$.

# B APPENDIX

Consider the following optimal control problem: it is required to maximize the functional

$$J(u(\cdot)) = \sigma(x(T)) - \int_0^T F^0(t, x(t), u(t)) dt,$$

over all $u(\cdot)$, where $x(\cdot)$ is the solution (Filippov, 1988, §1) of the differential equation

$$\frac{d}{dt} x(t) = F(t, x(t), u(t)), \quad t \in [0, T],$$

under the initial condition

$$x(0) = z.$$

Here $t$ is the time variable, $T > 0$ is the terminal instant of time, $x(t) \in \mathbb{R}^n$ is the current state vector, $u(t) \in U$ is the current control action vector forming the measurable function $u(\cdot)$, $U \subset \mathbb{R}^m$ is the nonempty compact set, $z \in \mathbb{R}^n$ is the fixed initial state vector, $F(t, x, u) \in \mathbb{R}^n$, $F^0(t, x, u) \in \mathbb{R}$, $(t, x, u) \in [0, T] \times \mathbb{R}^n \times \mathbb{U}$ are continuous with respect to $t$ and continuously differentiable with respect to $x$ functions, and $\sigma(x) \in \mathbb{R}$, $x \in \mathbb{R}^n$ is the continuous function. We assume that there exists a constant $c_{fg} > 0$ such that

$$\|F(t, x, u)\| \leq (1 + \|x\|) c_{fg}, \quad (t, x) \in [0, T] \times \mathbb{R}^n, \quad u \in U.$$

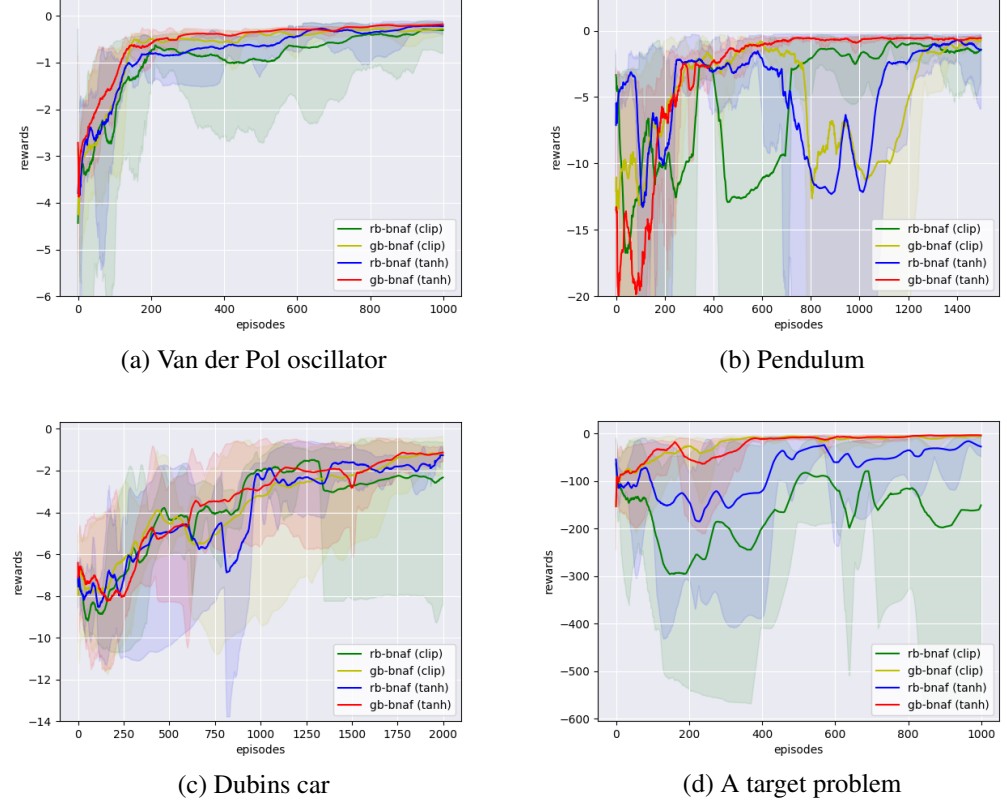

(a) Van der Pol oscillator

(b) Pendulum

(c) Dubins car

(d) A target problem

Figure 2: Results of RB-BNAF, GB-BNAF algorithms with clip and tangent functions averaged over 5 seeds.

Similar to (25), denote

$$\mathcal{H}(t, x, s, u) = \langle F(t, x, u)u, s \rangle - F^0(t, x, u).$$

**Theorem 4.** Let the value function $V_*(t, x)$ be continuously differentiable. Then, for every $\varepsilon > 0$, there exists $k_* > 0$ such that, for every $k \geq k_*$, the function

$$Q(t, x, u) = V_*(t, x) + A(t, x, u),$$

$$A(t, x, u) = \mathcal{H}(t, x, \nabla_x V_*(t, x), u)\Delta t_k - \max_{u' \in U} \mathcal{H}(t, x, \nabla_x V_*(t, x), u')\Delta t_k \qquad (36)$$

satisfies the inequality

$$\left| Q(t_i, x, u) + F^0(t_i, x, u)\Delta t_k - \max_{u' \in U} Q(t_{i+1}, x', u') \right| \leq \varepsilon \Delta t_k,$$

$$x' = x + F(t_i, x, u)\Delta t_k, \quad u \in U, \quad x \in S(t_i), \quad i \in \overline{0, k-1},$$

where we assume $Q(t_k, x', u') = \sigma(x')$.

**Proof of Theorem 4** can be obtained similar to the proof of Theorem 1.

For the family of q-function (36), based on Theorem 4, one can also prove the result similar to Theorem 2.

## C  APPENDIX

Figure 2 shows learning curves of RB-BNAF and GB-BNAF algorithms with the clip and hyperbolic tangent functions for the considered examples. Note that there are no cases in which algorithms using the clip functions show better performance comparing with the corresponding algorithms using the hyperbolic tangent function. Although the difference does not seem very significant.

