# OpenReview forum: "Continuous Deep Q-Learning in Optimal Control Problems: Normalized Advantage Functions Analysis"
_ICLR.cc/2022/Conference — ICLR 2022 Submitted_

### Official Review · Reviewer_J1Yy · 2021-10-29

**Correctness:** 4
**Technical Novelty And Significance:** 2
**Empirical Novelty And Significance:** 2
**Recommendation:** 3
**Confidence:** 4

**Main Review:**

The paper addresses an important class of optimal control problems in continuous action spaces that has been traditionally difficult to solve by general RL algorithms, due to the need to solve a non-trivial optimization problem to find the optimal action every time the value function is computed by means of a Bellman back-up. By assuming that the advantage term of the value function is quadratic in the action, this optimization problem can be solved much more efficiently, thus speeding up policy computation significantly. The previously proposed NAF algorithm makes this assumption, as do other algorithms, for example the iLQR algorithm, without necessarily knowing for sure whether this general assumption is correct or not. A question of practical significance then is to try to understand under what circumstances this assumption is justified. In this paper, the authors prove that if the cost functional of the optimal control problem contains a term that is quadratic in the control effort (action), then the approximation is reasonable, and the true value function of the decision problem can be approximated in this way arbitrarily well. This is a good result, because this kind of "minimum effort" problems with non-linear dynamics but costs quadratic in the control effort are very common in practice and of high practical significance.

Other than this proof, the contributions of the paper are rather limited. Several variations of the original NAF algorithm are proposed, and are experimentally compared on four reasonably difficult problems of the kind identified in the theoretical analysis. However, I do not see any overwhelming superiority of the new variants in comparison with the original NAF algorithm, except maybe on the Dubins car. At any rate, for practitioners of RL faced with a continuous control problem, the main question is probably not which of the NAF variants is best, but how these variants compare with other RL algorithms that do not leverage the knowledge that the value function must be quadratic in the control effort everywhere in the state space. Maybe a comparison with several such algorithms would better persuade practitioners to use the NAF algorithms and its variants?

Also, I am not entirely sure what this paper has to do with the main topic of the conference, learning representations. Maybe the way the value function is represented by means of the three neural networks for V, mu, and P is related to representations, but this representation is not learned, but rather deduced based on understanding of the structure of the cost functional of the problem. I like the paper in general, but it might be of more interest to the audience of a control conference than to that of ICML.

Some minor typos;
P.2, first paragraph: "whose dynamic, in fact, described" -> "whose dynamics are, in fact, described"
P.7, caption of Figure 1: "BNAD" -> "BNAF"?
P.7, first paragraph in 5.2: "learn their using" -> "learn their parameters using"?



**Summary Of The Paper:**

The paper investigates the applicability of quadratic approximations to the advantage term in the value function of RL problems in continuous action spaces, and the effectiveness of the previously proposed NAF algorithm for finding optimal policies for such problems. The main contribution is identifying a class of optimal control problems where the cost is a quadratic form of the control effort (commonly known as "minimum-effort problems"), and proving theoretically that the value function for such problems can be approximated arbitrarily well by the kind of quadratic approximations eteh mployed by the NAF algorithm.

**Summary Of The Review:**

The paper analyzes under what conditions the value function of an optimal control problem in continuous control spaces can be approximated by a quadratic function of the control effort, and prove that this approximation is reasonable when the cost functional itself includes a term that is quadratic to the control effort. Several modifications to the previously proposed NAF algorithm are described and tested, but it is not clear how much better or faster they learn in comparison with standard RL algorithms that do not make use of such quadratic approximations.

---

> ### Author Response · Authors · 2021-11-19
> **Response Letter**
>
> First of all I would like to thank you for the review. Let me comment on it below.
>
> “Other than this proof, the contributions of the paper are rather limited. Several variations of the original NAF algorithm are proposed, and are experimentally compared on four reasonably difficult problems of the kind identified in the theoretical analysis. However, I do not see any overwhelming superiority of the new variants in comparison with the original NAF algorithm, except maybe on the Dubins car.”
>
> We would also mention A target problem, because it is in this example that the GB-BNAF algorithm shows its greatest superiority.
>
> “At any rate, for practitioners of RL faced with a continuous control problem, the main question is probably not which of the NAF variants is best, but how these variants compare with other RL algorithms that do not leverage the knowledge that the value function must be quadratic in the control effort everywhere in the state space. Maybe a comparison with several such algorithms would better persuade practitioners to use the NAF algorithms and its variants?”
>
> You are right, we consider different modifications of NAF and compare them to only NAF. We did not consider other algorithms, because their comparison with NAF was presented earlier in paper.
>
> Shixiang Gu, Timothy Lillicrap, Ilya Sutskever, and Sergey Levine. Continuous deep q-learning with model-based acceleration. Proceedings of the 33rd International Conference on Machine Learning, 48:2829–2838, 2016.
>
> Now, we have added to our experiments DDPG algorithm as one of the most famous algorithms solving reinforcement learning problems.
>
> “Also, I am not entirely sure what this paper has to do with the main topic of the conference, learning representations. Maybe the way the value function is represented by means of the three neural networks for V, mu, and P is related to representations, but this representation is not learned, but rather deduced based on understanding of the structure of the cost function of the problem. I like the paper in general, but it might be of more interest to the audience of a control conference than to that of ICML.”
>
> Despite the fact that we consider only certain class of Reinforcement Learning problems, we think that our paper is closely related to the Reinforcement Learning topic. The ICLR conference has “representation learning for planning and reinforcement learning” topic, which, as we consider, is appropriate for similar papers. There were a lot of papers on the reinforcement Learning topic which had been published in ICLM. For example,
>
> Timothy P. Lillicrap, Jonathan J. Hunt, Alexander Pritzel, Nicolas Heess, Tom Erez, Yuval Tassa, David Silver, and Daan Wierstra. Continuous control with deep reinforcement learning. International Conference on Learning Representations (ICLR), 2016.
>
> which is closely related to the research area of our paper.
>
> “Some minor typos; P.2, first paragraph: "whose dynamic, in fact, described" -> "whose dynamics are, in fact, described" P.7, caption of Figure 1: "BNAD" -> "BNAF"? P.7, first paragraph in 5.2: "learn their using" -> "learn their parameters using"?”
>
> These types have been corrected.

---

> > ### Comment · Reviewer_J1Yy · 2021-11-24
> > **Response to authors' response**
> >
> > Thank you for making an effort to address my comments. The included comparison with DDPG, which makes this paper more self-contained. I also accept the argument that maybe ICLR is a suitable venue for the paper. As a result, I am willing to revise my rating upwards to 4, but this would likely not be sufficient to push the paper above the bar for acceptance, unless the other reviewers revise their ratings significantly upwards, too.

---

### Official Review · Reviewer_8xxB · 2021-11-02

**Correctness:** 3
**Technical Novelty And Significance:** 1
**Empirical Novelty And Significance:** Not applicable
**Recommendation:** 3
**Confidence:** 5

**Main Review:**

The manuscript cannot be accepted for publication in the current format. Some, but not all, shortcomings are briefly mentioned below.

First, the setting looks artificial and non-realistic, and the title is misleading, as the continuity is converted to discreteness! In Sec 2, the problem is not well explained, e.g., knowns and unknowns are not specified. Optimal policy is referred to as Greedy policy. I do not think that solving the problem in (1) is equivalent to those above (1), as the authors claim.

Further, since (6) is a classical noiseless optimal control problem and the goal is to find a possibly anticipative control signal, it is not clear how it falls under the umbrella of non-anticipative RL policies in Sec 2. Also importantly, it is unclear why the proposed method is better than the classical/existing solutions for (6). Note that it is assumed that the value function is known. So, directly solving HJB equation looks a better option, especially as it is exact and there is no approximation.

The proposed framework, by (11) and Thm 2, is applicable only to small T setting, which is not the common setting in control or RL. Putting this together to the other assumptions, the main claim of the paper is that Q-func is continuous with time. However, it seems like an immediate consequence of smoothness of the value function in Thm 1. Thm 3 is a counter example in fact, which is artificial since simply extending the U interval leads to letting \mu be included in the family of policies.

Overall, the impression of the reader is that the authors are not familiar enough with the frameworks of reinforcement learning and optimal control.

Some concepts and quantities are not defined, e.g., after (3). There are several issues with the presentation including grammatical and writing errors, typos, inaccuracies, and unclear statements:
"our improvements", "can act optimally", "solution the Bellman", "For the case ..descent", "Deep ...challenging tasks", "the presented paper", "a wider (in some sense) class", "whose dynamic", "a new family of quadratic", "an aim", "as a solution", (1), "during the learning", "buffer", "consists in", "as well as (see 5) quadratic", to name a few.


**Summary Of The Paper:**

The paper applies the method of quadratic approximation of Q-functions to a continuous-time optimal control problem by discretizing the time. It is shown that under technical assumptions, if the horizon is short and the discrete times are dense enough, the approximation becomes accurate.

**Summary Of The Review:**

The manuscript cannot be accepted for publication in the current format. First, the setting looks artificial and non-realistic and the problem is not well explained. Further, in (6), it is not clear how it falls under the umbrella of non-anticipative RL policies in Sec 2. Also importantly, it is unclear why the proposed method is better than the classical/existing solutions for (6). The proposed framework, by (11) and Thm 2, is applicable only to small T setting, which is not the common setting in control or RL. The main claim of the paper is that Q-func is continuous with time, which seems like an immediate consequence of smoothness of the value function in Thm 1. Thm 3 is a counter example, but is artificial since simply extending the U interval leads to letting \mu be included in the family of policies. Some concepts and quantities are not defined, and here are several issues with the presentation.

---

> ### Author Response · Authors · 2021-11-19
> **Response Letter**
>
> First of all I would like to thank you for the review. Let me comment on it below.
>
> 1. “First, the setting looks artificial and non-realistic, and the title is misleading, as the continuity is converted to discreteness!“
>
> “Continuous Deep Q-Learning” is the name of the algorithm which was introduced in
>
> Shixiang Gu, Timothy Lillicrap, Ilya Sutskever, and Sergey Levine. Continuous deep q-learning with model-based acceleration. Proceedings of the 33rd International Conference on Machine Learning, 48:2829–2838, 2016.
>
> Next, this name was used, for example, in papers
>
> Xingping Dong, Jianbing Shen, Wenguan Wang, Yu Liu, Ling Shao, and Fatih Porikli. Hyperparameter optimization for tracking with continuous deep q-learning. Proceedings of the IEEE Conference on Computer Vision and Pattern Recognition (CVPR), pp. 518–527, 2018. doi: 10.1109/CVPR.2018.00061.
>
> Junya Ikemoto and Toshimitsu Ushio. Continuous deep q-learning with simulator for stabilization of uncertain discrete-time systems. arXiv:2101.05640, 2021.
>
> The word “continuous” emphasizes that we consider problems with continuous action spaces. We do not use a discretization of the action space within our paper
>
> 2. “In Sec 2, the problem is not well explained, e.g., knowns and unknowns are not specified.”
>
> Section 2 contains a brief description of previous ideas and results concerning reinforcement learning theory and q-learning, deep q-learning, continuous deep q-learning  algorithms. Unfortunately, we do not have an opportunity to clarify it in more detail because of the paper size limit.
>
> 3. “Optimal policy is referred to as Greedy policy”
>
> An optimal policy is the policy which maximizes J(mu), but it is known that the greedy policy defined by the optimal q-function is optimal.
>
> 4. “I do not think that solving the problem in (1) is equivalent to those above (1), as the authors claim.”
>
> We do not claim the equivalence, but we show (see (14)-(17)) that discrete optimal control problem (12), (13) can be considered as MDP.
>
> 5. “Further, since (6) is a classical noiseless optimal control problem and the goal is to find a possibly anticipative control signal, it is not clear how it falls under the umbrella of non-anticipative RL policies in Sec 2”
>
> Despite the fact that, in problem (6)-(8), we need to find an action function u(.) maximizing J(mu), it is known (see chapter I, section 5 in M. Bardi and Italo Capuzzo Dolcetta. Optimal Control and Viscosity Solutions of HamiltonJacobi-Bellman Equations. Birkhauser, Boston, 1997) that this function can approximately be obtained as a result of an optimal feedback (non-anticipative) strategy. This strategy can be considered as the optimal policy in the corresponding MDP.
>
> 6. “Also importantly, it is unclear why the proposed method is better than the classical/existing solutions for (6). Note that it is assumed that the value function is known. So, directly solving HJB equation looks a better option, especially as it is exact and there is no approximation.”
>
> We do not assume that the value function is known. We investigate only the form of the q-function. Then, we use this form and neural network approximations to solve the Bellman equation approximately. If you mean finite difference methods as directly solving HJB equation methods, then these methods can be applied only for the small dimension problems.
>
> 7. The proposed framework, by (11) and Thm 2, is applicable only to small T setting, which is not the common setting in control or RL.
>
> We do not assume any limitation on the size of T. We do not also claim that we cover all reinforcement learning problems. We cover only the certain reinforcement learning problems which are described in (14)-(17).
>
> 8. “Putting this together to the other assumptions, the main claim of the paper is that Q-func is continuous with time. However, it seems like an immediate consequence of smoothness of the value function in Thm 1.”
>
> It is true that the function defined in (18) is continuous with respect to t. However, it is not the main result of our paper. Moreover, we do not use this fact for getting our results.
>
> 9. “Thm 3 is a counter example in fact, which is artificial since simply extending the U interval leads to letting \mu be included in the family of policies.”
>
> In most optimal control problems as well as Reinforcement learning problems, the set of admissible action values is determined by the problem statement. We can not extend this set.

---

### Official Review · Reviewer_6AfP · 2021-11-03

**Correctness:** 3
**Technical Novelty And Significance:** 2
**Empirical Novelty And Significance:** 2
**Recommendation:** 3
**Confidence:** 2

**Details Of Ethics Concerns:**

Ethics Concerns:
None

**Main Review:**

Strengths:
- Paper is relatively clear and organized, and it is easy to understand the high level idea of the paper.
- Great experiments, clear examples of how RL environment fall under the class of control problem

Weaknesses:
- The paper immediately jumps to a specific class of optimal control problem in Section 3. What is the intuition behind choosing such a class?
- Needs more seeds, unclear in Figure 1 if RB-BNAF is better than NAF. NAF also seems to have very high variance and hence needs more runs. (Please do at least 5 seeds for all other baselines, if not 10 seeds for NAF)
- In section 4, the new class of Q functions seems to be arbitrarily proposed. I believe this is the biggest contribution of the paper, so I recommend making this class of functions more intuitive (perhaps explain in Appendix how this class of functions are conceived of).
- Should ablate on what happens when there is no bounding of Tanh for BNAF.

Questions:
- What are the limitations to this approach (beyond limited to a certain class of problems)?
- What is future work (e.g. extending this to more general classes of problems)?

**Summary Of The Paper:**

-The authors prove that a discrete approximation of a class of optimal control problems can be recasted as an RL MDP.
-The authors prove that the original NAF formulation of the Q-function cannot approximately solve the MDP defined above, and hence propose a new quadratic formulation of the Q-value. They apply their new formulation via Bounded, Reward-based, and Gradient-based BNAF.
-The authors evaluate their proposed agent over 4 optimal control environments.

**Summary Of The Review:**

Overall:
Based on my limited understandings and mathematical background, the authors propose modifications to NAF, prior work, and prove it can solve certain classes of control theory problems.

---

> ### Author Response · Authors · 2021-11-19
> **Response Letter**
>
> First of all I would like to thank you for the review. Let me comment on it below.
>
> In Appendix B, we have considered the more general class of optimal control problems and have proved results for this class similar to Theorems 1 and 2. The main goal of it is to show a direct link between optimal control problem parameters and the suitable form of the function A(t,x,u) (see (36)). We hope it answers some of your questions. In particular,
>
> “The paper immediately jumps to a specific class of optimal control problem in Section 3. What is the intuition behind choosing such a class?”
>
> The choice of the class (6)-(8) follows from (36) and our wish that the function A(t,x,u) is quadratic with respect to u and has a quite simple form.
>
> “In section 4, the new class of Q functions seems to be arbitrarily proposed. I believe this is the biggest contribution of the paper, so I recommend making this class of functions more intuitive (perhaps explain in Appendix how this class of functions are conceived of).”
>
> Probably, equality (36) can clarify, in a sense, an idea of the form of the function Q(t,x,u) for the class of optimal control problems (6)-(8).
>
> “What are the limitations to this approach (beyond limited to a certain class of problems)?”
>
> It seems that Theorem 4 extends our approach to the wider class of optimal control problems from a theoretical point of view. However, this extension implies a more difficult form of the function A(t,x,u) that is worse from a practical point of view. All these comments have been added to the paper after Theorems 1 and 2.
>
> “Needs more seeds, unclear in Figure 1 if RB-BNAF is better than NAF. NAF also seems to have very high variance and hence needs more runs. (Please do at least 5 seeds for all other baselines, if not 10 seeds for NAF)”
>
> We have conducted experiments with 5 seeds and have updated the figure.
>
> “Should ablate on what happens when there is no bounding of Tanh for BNAF.”
>
> We have conducted experiments to compare BNAF, RB-BNAF, GB-BNAF algorithms for the clip and tanh functions. The difference is not too significant, but it exists. We have shown it for RB-BNAF, GB-BNAF algorithms in Figure 2 (see Appendix C.)
>
> “What is future work (e.g. extending this to more general classes of problems)?”
>
> In the future work, we hope to apply this approach to multi-agent reinforcement learning problems.

---

> > ### Comment · Reviewer_6AfP · 2021-11-30
> > **Response to Rebuttal**
> >
> > Thanks for addressing the majority of my concerns regarding this paper. Even with the paper's improvements, I think there is still a lot of work to be done before this paper can be accepted to ICLR (such as more complex environments). Hence, the rating will stay the same.

---

### Decision · Program_Chairs · 2022-01-20

**Decision:**

Reject

**Comment:**

The paper concerns learned Q functions in continuous action spaces wherein the action-value function is assumed to be quadratic in the action variable, and thus can be maximized in closed form. The paper identifies a class of optimal control problems for which the approximation is reasonable and produces a proof to this effect.

Reviewers found the manuscript clear (6AfP). Reviewer J1Yy notes that the main result of the paper is useful and good to have, as "problems with non-linear dynamics but costs quadratic in the control effort are very common in practice and of high practical significance". On the negative side, reviewer J1Yy considered the contribution beyond the central proof rather light and the empirical study inconclusive and questioned the appropriateness of the venue; a comparison to DDPG was added and while a convincing argument was made as to ICLR's appropraiteness, J1Yy was not willing to move their score beyond a 4 (it does not seem the upward adjustment was actually made). 6AfP noted concerns with the presentation and number of seeds, though their concerns seem to have been addressed in an update. 8xxB was the paper's most ardent critic, who found fault with the presentation (starting with the title). The core of 8xxB's criticism seems to be that by narrowing the scope of problems considered, we are left with a problem domain that is already well solved by classical control, as well as contention about the use of "continuous". The authors offered a thorough rebuttal but the authors and 8xxB were unfortunately unable to see eye to eye on these issues. 8xxB requested more explanation, though a request by the authors to specify the precise scope of what further was required went unanswered.

The AC's own reading of the paper matches J1Yy's assessment most closely. There is a contribution here, in the form of connecting previously proposed RL algorithms for continuous action spaces and discretized time to the literature on optimal control, and exploring cases that match NAF's inductive assumptions, but agree that the contribution is of a rather limited nature. I also believe that the paper has improved through the feedback of reviewers J1Yy and 6AfP. I hesitate to recommend acceptance given a universally negative appraisal and in particular the fact that 8xxB was not satisfied in the end. I hope the authors will continue to improve the manuscript with a more thorough empirical interrogation and adjustments in presentation in light of 8xxB and 6AfP's feedback.